# Intracranial Venous Sinus Stenting in Idiopathic Intracranial Hypertension: A Case Report and Review of the Literature

**DOI:** 10.3390/brainsci11030382

**Published:** 2021-03-17

**Authors:** Dinesh Ramanathan, Zachary D. Travis, Emmanuel Omosor, Taylor Wilson, Nikhil Sahasrabudhe, Anish Sen

**Affiliations:** 1Department of Neurosurgery, Loma Linda University Medical Center, Loma Linda, CA 92354, USA; TAYWilson@llu.edu (T.W.); NSahasrabudhe@llu.edu (N.S.); ASen@llu.edu (A.S.); 2School of Medicine, Loma Linda University, Loma Linda, CA 92350, USA; zacharydtravis@gmail.com (Z.D.T.); eomosor@llu.edu (E.O.)

**Keywords:** venous stenting, arachnoid cyst, idiopathic intracranial hypertension

## Abstract

We describe a case of severe headaches, double vision, and progressive vision loss secondary to a ruptured intracranial cyst (IAC) in a 31-year-old woman with no relevant past medical history. The case is peculiar because drainage of the subdural hygroma led to a minimal improvement in vision with persistent elevated intracranial pressure (ICP). Further exploration revealed transverse sinus stenosis necessitating stenting. Evaluation post-stenting showed marked reduction of ICP and improvement in symptoms. This report underscores the importance of comprehensive work-up and suspicion of multiple underlying etiologies that may be crucial to complete resolution of presenting symptoms in some cases. We provide an overview of the clinical indications and evidence for venous sinus stenting in treating idiopathic intracranial hypertension (IIH).

## 1. Introduction

Occam’s razor, or the law of parsimony, is the principle that “entities should not be multiplied without necessity”, and when applied to medicine, that a single diagnosis generally links all symptoms. Since students, Occam’s razor is repeatedly discussed because the vast majority of times it is correct; however, when correction of one structural cause of symptoms does not correct the other symptoms, it is critical to look for additional potential causes, which may or may not have been initially obvious. In this manuscript, we present an interesting case a of patient with two concurrent structural causes of headache, which is in contrast to the usual thinking according to Occam’s razor.

## 2. Case History

A 31-year-old female with no significant past medical history presented with 4 weeks of progressive bilateral vision loss, headache, and diplopia. She was initially diagnosed with migraines, but her symptoms persisted, prompting a visit to the Emergency Department. On physical examination, the patient’s right eye had a lateral gaze palsy and perceived motion only, whereas her left pupil was nonreactive with impaired color and light perception in that eye. No other neurological deficits were appreciated. Imaging studies demonstrated a left anterior temporal pole arachnoid cyst with a 19 mm left subdural hygroma with mass effect causing 4 mm of left to right midline shift, most concerning for a ruptured arachnoid cyst (Figure A1a).

The patient was taken to the operating room for a left frontal burr hole for drainage of her subdural hygroma. Post-operatively, she continued to have headache and minimal improvement in light perception bilaterally. Lumbar puncture demonstrated elevated intracranial pressure (ICP) with an opening pressure of 30.4 cmH_2_O consistent with idiopathic intracranial hypertension; magnetic resonance (MR) brain and MR venogram showed left transverse sinus stenosis (Figure A1b). Subsequently, a right frontal external drain (EVD) was placed for ICP management, after which there was significant improvement in the patient’s vision.

Diagnostic cerebral angiogram with venous manometry was then performed measuring 39 cmH_2_O in her left transverse sinus and 16 cmH_2_O in her left sigmoid sinus (Figure A2). The pressure gradient of 23 cmH_2_O supports the diagnosis of idiopathic intracranial hypertension with left sinus stenosis, and she subsequently underwent left transverse sinus with a post-stenting pressure gradient of 7 cmH_2_O. After the procedure, her headache improved significantly, and she was able to count her fingers using both eyes. Within 24 h, she also had near complete resolution of her right lateral gaze palsy. She was eventually discharged home on a 6-month course of aspirin and clopidogrel. At 1 month follow-up, the patient continued to show improvement in her visual acuity.

## 3. Discussion

We present a unique case of a patient with headaches and progressive visual loss secondary to two etiologies—idiopathic intracranial hypertension (IIH) and associated ruptured arachnoid cyst. It was critical to evaluate and treat for both causes to relieve the patient’s symptoms. Headaches are a common symptom and, thus, if headache, and in this case visual loss, does not improve after treatment of a seemingly obvious structural pathology of subdural hygroma, then additional differential diagnoses should be explored.

Our patient presented with an obvious structural finding of a subdural hygroma with evidence of a mass effect on surrounding brain structures, with concern for associated increased ICP. Despite drainage of the subdural hygroma, her headache and visual symptoms persisted, prompting additional work up with MRI brain and MR venogram revealing left transverse sinus stenosis consistent with an underlying diagnosis of IIH.

Few reports in the literature have reported IIH developing after successful treatment of intracranial cyst (IAC), in some cases after up to several years [1,2,3,4]. One paper reported a series of three cases in which the IIH symptoms became clinically apparent or evolved after the surgical treatment of arachnoid cyst [2]. Similarly, another paper based on the clinical presentation and treatment of three arachnoid cyst patients, and later IIH, described a mechanism explaining arachnoid cyst formation and elevation of ICPs [3]. A third paper described four cases of surgically treated IAC who developed persistent IIH, of which three patients continued to be symptomatic with severe headaches [4]. They proposed a disequilibrium of cerebrospinal fluid (CSF) hydrodynamics as a probable basis of onset or exacerbation of IIH after IAC treatment. Diversion of cystic and subdural CSF into basal cisterns decompensating a tenuous CSF circulatory mechanism has been proposed [2,5]. The possibility of cyst fluid secreted by cyst lining cells being different than CSF, potentially leading to problems in CSF dynamics or circulation, has also been raised as possible mechanism of developing IIH [6]. Other investigators were not able to rule out a mere coincidence of IAC and IIH, and recommend close evaluation of imaging details and history to optimally address all underlying etiologies towards symptom resolution [7].

IAC are congenital fluid-filled malformations in subarachnoid spaces/cisterns and major cerebral fissures. Despite an estimated prevalence of 1.4%, little is known about the pathogenesis of these presumed developmental anomalies of the arachnoid [8]. Richard Bright first reported medical cases in 1831 involving IAC. These were found in patients from all age groups but mostly (75%) occurred in children [9]. The natural history of arachnoid cysts is not clear. The majority of the arachnoid cysts remain stable in size, although some can enlarge secondarily to CSF secretion from the cyst membrane, fluid osmosis mediated by higher protein cystic contents, or mechanical entrapment due to the ball-valve mechanism [7,10,11]. The presence of arachnoid cysts in settings of developmental abnormalities, trauma, and infections point to several possible etiologies although it is most commonly developmental. It is thought that IAC form due to CSF flow disturbances in the early phase of subarachnoid space formation secondary to leptomeningeal maldevelopment [4,11,12,13]. The cyst wall histology is generally reflective of its anatomic location. Although the posterior fossa cysts display choroid plexus remnants, suprasellar and prepontine cysts harbor neuroglial elements [7]. The structural features of the arachnoid cyst wall that distinguish it from the normal arachnoid membrane are as follows: (1) splitting of the arachnoid membrane at the margin of the cyst; (2) a very thick layer of collagen in the cyst wall; (3) the absence of traversing trabecular processes within the cyst; and (4) the presence of hyperplastic arachnoid cells in the cyst wall, which presumably participate in collagen synthesis [11]. Most small IAC are asymptomatic and do not require surgery. However, larger cysts can have a local mass effect on neurovascular structures, leading to neurological symptoms which warrant surgery [14]. Symptoms may include headaches/elevated ICP, hydrocephalus, or cystic rupture into the subdural space leading to subdural CSF hygroma or hemorrhage. There is no strong consensus on the pathophysiology of IAC. Some theories suggest that IAC arise from a congenital splitting of arachnoid membrane layers during fetal development, resulting in connected CSF entrapment and/or accumulation in this “potential space.” [9].

IIH is a disorder secondary to persistent elevated ICP without an otherwise distinguishable pathology [15]. First described by German physician Heinrich Quincke, it was later labeled benign intracranial hypertension by Foley [15,16]. The diagnosis is based on the classic triad of chronic headaches, papilledema, and the absence of structural pathologies such as intracranial lesions or ventriculomegaly. Other symptoms associated with IIH include tinnitus and progressive vision loss, and diplopia associated with abducens palsy can be seen occasionally [16]. Long term papilledema can lead to progressive loss in the visual field (enlargement of physiologic blind spot) and visual acuity.

Although the exact pathophysiology remains debated, proposed etiologies of IIH include excessive secretion of CSF and/or increased resistance to CSF absorption, primarily centered around decreased venous outflow. IIH can be categorized as primary or secondary. Primary IIH is truly idiopathic without a known etiology. Primary IIH typically occurs in overweight females of reproductive age, although it is important to remember that it is not limited to this demographic. Secondary IIH is not truly idiopathic as it results from an identifiable underlying etiology. These etiologies are vast, including metabolic or medication-induced (tetracycline, lithium, excessive ingestion of vitamin A or vitamin A-containing medications, and chronic steroid use), structural (dural sinus stenosis, most commonly transverse sinus), infection (meningitis), or secondary to systemic disease states, particularly those associated with chronic inflammation. Our patient had a case of secondary IIH that we were able to identify and successfully treat.

Initial angiograms performed demonstrated bilateral or unilateral transverse sinus stenosis [18]. Higgins et al. described the first venous sinus stenting (VSS) for refractory IIH and significant improvement of trans-stenosis pressure gradient and symptomatic improvement. However, the relationship between venous stenosis and IIH is controversial. It is not entirely clear if venous stenosis is a cause of IIH or merely a manifestation of IIH [19]. Buell et al. published in vivo evidence of venous stenosis as the manifestation of IIH, providing lumbar puncture findings with simultaneous measurement of trans-stenosis pressure gradient and venous stenosis. CSF drainage was noted to improve the trans-stenosis pressure gradient and venous stenosis [20]. Regardless, the procedure can be effective in preventing the positive feedback loop and breaking the starling-like resistor effect [21]. By facilitating the distal cerebral outflow, the trans-arachnoid villi gradient can increase, leading to improved CSF resorption [15]. Appropriate preoperative workup for VSS includes visual field testing, papilledema evaluation, ICP measurement (via ICP monitor or lumbar puncture), and MR venography/cerebral angiography for venous manometry. Pressure is measured in the superior sagittal sinus (SSS), bilateral transverse sinuses, sigmoid sinus, and cervical internal jugular. A trans-stenotic pressure gradient of >8 cmH_2_O is considered an indication for venous stenting. Antiplatelet medication (aspirin 325 and clopidogrel 75 mg daily) is prescribed for 1 week before VSS to keep the P2Y_12_ response unit (PRU) and aspirin response units (ARU) levels in the optimal range. Generally, patients are placed on dual antiplatelet therapy for 1 to 6 months and aspirin 325 for a further 3 months to lifetime use [15]. Several large case series have demonstrated a high success rates for headache symptom relief after VSS [15,22,23,24]. One paper reported improvement in headache severity in about 80% of patients, and stenosis resolution in the stented segment in 100% of patients over a 2.5 year follow-up, with no complication noted [22]. Cerebellar hemorrhage could be a possible complication of VSS, and the risk is inversely associated with age of the patient [25].

Patients presenting with IIH after surgical treatment of an arachnoid cyst have been reported in the literature. No definitive causative mechanism is known yet to definitively associate these diagnoses although alteration or disequilibrium of CSF hydrodynamics is tentatively implicated as a putative etiology. Early work-up of all potential differentials and treating the underlying pathogenesis is essential to achieving headache resolution and vision preservation in such cases.

## 4. Conclusions

Although the majority of the time Occam’s razor holds true, when all symptoms do not improve after treatment of one structural pathology, it is paramount to consider other differential diagnoses, even if these other diagnoses may not be initially apparent.

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
