# Peer review of "Intracranial Venous Sinus Stenting in Idiopathic Intracranial Hypertension: A Case Report and Review of the Literature"

_brainsci, 2021, doi:10.3390/brainsci11030382_

Round 1

Reviewer 1 Report

The authors provide an overview to the current treatment options and pathophysiological concepts of IAC and IIH, by linking it to a single case report suffering from both pathologies.

The manuscript is well written, the case is well presented and comprehensive to the readership.

However the discussion and presentation of the current knowledge and literature has to be expanded.

  • The authors should invest some sentences in diagnosis and follow up of IIH, e.g. trans orbital ultrasound. Moreover not all IIH are due to sinus stenosis. The etiologies could be put  in a table with treatment options and followup procedures (thrombosis of sinus; stenosis, idiopathic/therapeutic options for each, such as medication (OAC or acetazolamide...), body weight loss etc. ).

If the authors manage to deepen and expand the discussion the value of the manuscript can be improved. In its current form it is not adding too much information to what is already known and does not serve as guidance for therapeutic or diagnostic dilemma. 

Author Response

Thank you for your very constructive review. We believe that your suggestions help guide this manuscript towards the intention that we, as a team, desire. We are working on incorporating a diagnosis and treatment algorithm into the discussion, that has not quite been completed yet due to time restraints. We are submitting what we have so far based on the second reviewer's suggestions and would like to request a week's extension to finalize the edits to your suggestion.

Reviewer 2 Report

The authors report a case of elevated intracranial pressure with headache from a ruptured arachnoid cyst with subdural fluid collection and transverse sinus stenosis. The two conditions appeared to be unrelated. The authors treated the subdural collection with drainage and the transverse sinus stenosis with a stent. The patient improved.

The case is interesting in that the patient had two concurrent structural causes of headache, in contrast to the usual Occam's razor view that a single diagnosis will generally link conditions together. This deviation from the usual medical thinking should actually be the main thrust of the manuscript and provides an appropriate conclusion - that when correction of a structural cause of symptoms does not resolve the symptoms, one should look for other potential causes which may not have been obvious or suspected.

In its current form, the manuscript is very wordy. The INTRODUCTION should better indicate the objective of the case report. The CONCLUSIONS should be briefer and more to the point as started above. The authors should provide an explanation as to why they only drained the subdural collection and did not fenestrate the arachnoid cyst. More details about the stenting procedure - what device, how placed, etc., would be helpful.

Author Response

Thank you for your very thorough review of this manuscript. We really appreciate the ability to improve on our work. So far we have made some edits to the introduction and the conclusion to tie in the brilliant point you raised about Occam's razor. We are still working on cutting down some bulk in the discussion to make it less wordy and also address your question as to why we did not fenestrate the cyst. Time constraints have plagued our ability to finish the edits on our end. We will upload what we have so far but would like to request a one-week extension to finalize the edits to your suggestion. Thank you. 

Reviewer 3 Report

This manuscript covers a very important clinical topic which is of interest to acute care clinicians. There are a few minor edits suggested which can make this paper stronger. 

Some of the images appear slightly blurred and may need a higher resolution to improve clarity. 

Otherwise, good paper. Thanks for sharing your work. 

Author Response

Thank you very much for your constructive comments. We are working on getting higher-quality images to replace the old ones and have made the suggested edits to the manuscript. Thank you.

Round 2

Reviewer 2 Report

The authors have addressed the reviewer's comments satisfactorily. The manuscript needs some English language editing.